# High throughput machine learning pipeline to characterize larval zebrafish motor behavior

John Hageter[1]☯, John Efromson[2]☯, Brooke Alban[1], Audrey DelGaudio[1], Veton Saliu[2], Monica Wassef[2], Mark Harfouche[2], Eric J. Horstick [1,3]*

1 West Virginia University, Department of Biology, Morgantown, West Virginia, United States of America, 2 Ramona Optics Inc., Durham, North Carolina, United States of America, 3 West Virginia University School of Medicine, Department of Neuroscience, West Virginia University, Morgantown, West Virginia, United States of America

☯ These authors contributed equally to this work.
* eric.horstick@mail.wvu.edu

## Abstract

Using machine learning, we developed models that rigorously detect and classify larval zebrafish spontaneous and stimulus-evoked behaviors in various well plate formats. Zebrafish are an ideal model system for investigating the neural substrates underlying behavior due to their simple nervous system and well-documented responses to environmental stimuli. To track movement, we utilized an 8 key point pose estimation model, allowing precise capture of zebrafish kinematics. Using this kinematic data, we trained two random forest classifiers in a semi-supervised learning framework to classify various discreet behavioral outputs including stationary, scoot, turn, acoustic-startle like behavior, and visual-startle like behavior. The classifiers were trained on a manually labeled dataset, and their accuracy was validated showing high precision. To validate our machine learning models, we analyzed behavioral outputs during various stimulus evoked responses and during spontaneous behavior. For additional validation, and to show the utility of our recording and analysis pipeline, we investigated the locomotor effects of several established drugs with well-defined impacts on neurophysiology. Here we show that machine learning model development, enabled by semi-supervised learning developed classification models, provide detailed insights into the behavioral phenotypes of zebrafish, offering a powerful, high throughput method for studying neural control of behavior.

## Introduction

Animals navigate in their environment and respond to stimuli with distinct motor responses. The ability to record and quantify distinct motor responses can provide insight into sensorimotor processing, internal state, and neural circuit function. For example, limb or leg movement patterns can indicate grooming/cleaning in mice and *Drosophila*, respectively [1–3]. Similarly, in different *Drosophila* species, wing and

**Data availability statement:** Example data for each of the 96- and 24-well plate data for both the tap a dark flash stimulus along with behavior classification models and classifier training datasets are made available at Mendeley Data: DOI: 10.17632/sxvd6m53rf.1. Code for analysis is available open source at Gitlab: https://gitlab.com/ramona-applications/zebrafish-behavior.

**Funding:** This work was supported by National Science Foundation cooperative agreement OIA2242771, National Eye Institute R15EY036226, National Institute of General Medicine P20GM144230 awarded to Eric Horstick and the Office Of The Director, National Institutes Of Health under Award Number R44OD036187; the National Institute of Mental Health under Award Number R43MH133521; the National Institute of Environmental Health Sciences under Award Number R43ES036389; and the National Cancer Institute under Award Number R44CA285197 awarded to Ramona Optics. The content is solely the responsibility of the authors and does not necessarily represent the official views of the National Institutes of Health. Computational resources were provided by the WVU Research Computing Thorny Flat High Performance Computing cluster, which is funded in part by NSF OAC-1726534. The funders had no role in study design, data collection and analysis, decision to publish, or preparation of the manuscript.

**Competing interests:** The authors declare the following financial and personal relationships that may be considered as potential competing interests: J.E., V.S., M.W, and M.H. have a financial interest in Ramona Optics, Inc. This does not alter our adherence to PLOS ONE policies on sharing data and materials.

body movements indicate courtship behavior and can even be used to distinguish species [4–6]. Rates of startle initiation, which are highly conserved in evolution, can indicate arousal state, habituation, and model disease-like conditions [7–15]. Even in animals with arguably simple nervous systems, distinct motor responses delineate behavioral and neural context, such as in *C. elegans* where unique foraging states are delineated by use of specific motor responses [16–18]. These examples show that being able to accurately resolve distinct locomotor actions is a powerful tool to interrogate brain function and behavior.

Larval zebrafish are a powerful model for leveraging behavior to understand the underlying neural mechanisms [19–23]. A key advantage is that larvae move in distinct motor bouts and utilize a relatively small set of kinematically distinguishable motor features [24–27]. Prior work has described that larvae predominately navigate and respond to environmental cues with scoots, routine turns (R-turns), O-bends, and C-starts [28–31]. The utilization of these motor elements can describe general navigation and search within the environment (scoots and R-turns) [24,32,33], light flash induced startle (O-bend) [34,35], and rapid auditory startle (C-starts) [36–40]. Examining these motor responses has been crucial for using larval zebrafish to understand circuit mechanisms for pre-pulse inhibition, habituation, and neural plasticity [40–46]. Moreover, characterizing changes in the patterns of motor outputs in larval zebrafish has yielded new insights in understanding mechanisms underlying diverse and consequential disorders such as autism-spectrum disorders, Fragile-X, schizophrenia, and attention deficit disorders [42,47,48]. Another advantage of larval zebrafish is that their high fecundity can allow high-throughput analysis. Many studies have taken advantage of this high fecundity to successfully use zebrafish for high throughput assays, yet analysis typically relies on straightforward kinematic features, such as displacement or similar readouts for active versus inactive states [31,41,49,50]. Conversely, applying descriptive motor calls, e.g., scoot, C-start, etc., is typically associated with assays that record fewer larvae due to limitation in imaging throughput with conventional single objective microscopes and cameras [24,51,52]. Indeed, to integrate high throughout recording and analysis pipelines with rigorous classification of specific motor outputs poses several technical and computational hurdles due to the need for high speed and high-resolution imaging. Moreover, the ability to acquire high throughput data demands methods for efficient analysis, as manual analysis becomes burdensome and increasingly can add subjectivity or other confounds to an analysis. Therefore, expanding analytic tools available for high throughput behavior recording and analysis of detailed motor outputs can be a valuable resource.

Here, we generated an analytical machine learning pipeline to integrate with and complement the Ramona Optics Kestrel Multi camera array microscope's (MCAM) high speed, high resolution, and pose estimation software [53,54]. Machine learning has become a mainstay and transformative tool for computational neuroscience, and many other fields where it can fit models to data for diverse applications [29,55–57]. The rationale for the selected imaging platform was three-fold as it 1) provided the ability to record larval zebrafish in 24 or 96-well formats, which we propose offers a relatively high throughput method, 2) recorded images at speeds and resolutions we

predicted would allow development of accurate motor call models, and 3) provided complete access to raw data output through the usage of easily manageable NetCDF4 files. The MCAM platform utilizes multiple coordinated cameras coupled with analysis software that captures and reliably tracks individual larvae body kinematics extracted from 8 localized key points spanning the larval zebrafish body, originally based on DeepLabCut's pose estimation algorithm [58]. This automated positional pose estimation and software can extract generic kinematic features like displacement, speed, and heading direction metrics. However, these metrics alone do not allow for the determination of the specific motor outputs utilized by the larvae. To apply motor calls, we used an unsupervised clustering algorithm to efficiently label a large subset of our data followed by a supervised machine learning algorithm that we trained and evaluated with manually reviewed specific behaviors. Our training enabled models to reliably detect these four predominant motor outputs, in addition to stationary (non-moving) behavior. We further verify this analysis pipeline using established activity modulating pharmacological treatments, which we show can differentially impact motor responsiveness and utilization of distinct motor outputs. Altogether, we developed and verified motor behavior classification code that leverages high throughput imaging hardware, thus developing new open-source resources to study behavior and brain function using the larval zebrafish model. This study develops resources and an analysis pipeline for high-throughput data acquisition coupled with rapid analysis that can experimentally interrogate big data to address complex questions using larval zebrafish.

## Materials and methods

### Zebrafish husbandry

All the experiments were performed under the approval from the West Virginia University Institutional Animal Care and Use Committee (IACUC). Tübingen long-fin (TL) wildtype zebrafish (*Danio rerio*) were used for all experiments. All pharmacologic manipulations and behavior tracking was performed prior to or at 7 days post fertilization (dpf). All experiments are composed of at least 3 independent cohorts of zebrafish. Larval zebrafish were maintained on a 14/10-hour light-dark cycle and raised in E3h embryo media (5mM NaCl, 0.17mM KCl, 0.33mM $CaCl_2$, 0.33mM $MgSO_4$, and 1mM HEPES (pH 7.3)). Larvae were raised at a density of no more than 40 individuals per 30mL E3h. Larvae were raised, and behavior tested at 28ºC. Methods were included and approved in our IACUC protocol for humane euthanasia, anesthesia/analgesic use, and efforts to alleviate suffering. As per our approved protocol, the anesthesia/analgesic used was tricaine (MS222). Low dose tricaine was used as an anesthetic to minimize stress during handling, whereas high dose exposure was used for humane euthanasia.

### Behavior tracking

All behavior tracking experiments were performed on 5–7 dpf larvae using the Kestrel multi-camera array microscope (MCAM™, Ramona Optics Inc., Durham, NC). All larvae were acclimated to the behavior testing room for >30 minutes prior to recording and all recordings began with a 5-minute waiting period for larvae to acclimate to the behavior testing chamber. For all tracking assays, background illumination was set the same intensity used for rearing (80uW/cm² or 700 lux). For acoustic response tracking, larvae were recorded at 160 frames per second (fps) for 10 seconds and exposed to a tap stimulus after 5 seconds. This stimulation series was repeated for 6 acquisitions separated by 120 seconds. The tap module used with the Kestrel MCAM has four solenoids, two on each short side of the well plate, which are energized by the MCAM acquisition software at specified times during the video acquisition. To measure the intensity of the plate tapper module an accelerometer was placed in the center of a 96 well plate with 100uL of water in each well and the tap was quantified along three axes, x, y, and z. Peak acceleration in units of "g" are reported. The tap module exerted a force on the well plate so that the well plate experienced a peak acceleration of 3.78 g's. For the photic response tracking assay, larvae were recorded at 160 fps for 10 seconds. At 5 seconds, larvae were exposed to a light-off stimulus for 2 seconds before returning to normal illumination. For spontaneous behavior tracking, larvae were tracked in 24 and 96 well plates



at 160 fps. Larvae were recorded for 60 seconds separated by 180 seconds repeated 10 times for a total of 10 minutes of tracking.

### Data cleaning and egocentric alignment

For all experiments, 10 second acquisitions at 160 frames per second result in 1600 images each. Pose estimation data for each frame consists of eight y, x coordinates, one for each of the inferred key points, in an array of shape (8, 2). Individual frames' pose coordinates were concatenated into overlapping 40 frame windows so that each unit behavior window is an array of cartesian coordinates of shape (40 timepoints, 8 key points, 2 coordinates). For each 10 second acquisition this results in an array of behavior windows of shape (1560 windows, 40 timepoints, 8 key points, 2 coordinates) where there are 1,560 windows because the 1,561st frame only has 39 data points in the window and thus cannot be rationalized into a behavior unit of shape (40 windows, 8 key points, 2 coordinates).

Each behavior window of the 1,560-frame array was egocentrically aligned to move the coordinates from the image frame of reference to the context of the fish. To accomplish this a rotation was computed for the first frame of each window which would rotate the zebrafish so that the vector from center key point to snout key point is pointing straight up in the well image followed by a translation that moves the center key point to the center of the image. All other key points are translated the same as the center point. The first frame of each behavior window is oriented straight up in the well and its center point is in the center of the image. The same rotation and translation is applied to the following 39 frames in the behavior window. All pose estimation behavior windows have the zebrafish begin from the center of the image with the zebrafish pointed straight up and move for 40 frames.

All coordinate values in behavior windows were normalized by dividing by the maximum value and subtracting the mean value of the overall datasets. These values were different for 96-well and 24-well plates because the 24-well plate is physically bigger than 96- and thus have a larger spread of coordinate values. Behavior windows were flattened to a 1-dimensional feature vector of shape (1, 640) so that the overall acquisition feature array for each well has a shape of (1560, 640) with dimensions (frame_number, feature).

### Unsupervised model training

All experimental data was gathered and processed to flattened behavior windows as described and then clustered using KMeans clustering with k = 9 [59]. This resulted in a KMeans model where two clusters correspond to stationary behaviors and the other seven correspond to movement. This model was used to identify discrete swim bouts in zebrafish.

### Supervised model training

Once discrete motor bouts were isolated from the unsupervised model, forty frame windows were exported as.mp4 video clips. Clips were exported during movement events and during stationary periods. All video clips were reviewed manually before being given a class label. 96-well plates were given class labels "stationary", "movement", "AsLB", and "VsLB" while 24-well plates were given class labels "stationary", "scoot", "turn", "AsLB", and "VsLB". Stationary frames were found quickly by exporting windows identified as stationary by the KMeans model. Acoustic-like and visual-like windows were found by identifying movement clips near the stimulus window (frame 800 halfway through each 1600 frame acquisition) and then manually reviewing. Acoustic-like were enriched in tap stimulus datasets while visual-like were enriched in dark-flash datasets. "Movement" for 96-well and both "scoot" and "turn" classes were found by manually reviewing clips identified as movement by the KMeans model but not during the stimulus window. Manually reviewed video clips and associated pose estimation data were organized into folders marking each class for both 24- and 96-well plate datasets.

Each dataset (24- and 96-well datasets) were partitioned into training, and test data subsets for model training with proportions 90%/10%. Each training dataset was projected to a lower dimensional space using principle component analysis

(PCA) [59,60] with random state 2023, no inherent data whitening, and retaining 95% of the components representing dataset variability (0.95 components). The PCA transformation implemented on the training set was saved for future use. The resulting PCA transformed datasets were used to train two random forest classifiers (RFC) [61], one for each well plate configuration. The 96-well plate random forest infers 4 behaviors while the 24-well model infers 5 behaviors.

Inference is run by first preparing windowed behavior datasets of pose estimation data, egocentrically aligning and then flattening to feature vectors for each frame. Feature vectors are normalized and transformed using the saved PCA transformations and then inferred using the random forest classifiers. Machine learning pipelines were evaluated comparing the manual labels of the test set behavior windows to those inferred. The test set was not seen by either the PCA algorithm or RFC models prior to evaluation. Confusion matrices were plotted to visualize the model accuracy for each class and F1 score was computed for each model.

### Statistical analysis

All statistical analysis was completed in RStudio using package rstatix [62]. All t-tests are two tailed and assume equal variance among samples. Statistical significance was determined based on a p value being less than 0.05. Data in figures represents mean ± standard error of the mean. Normality was tested using the Shapiro-Wilks method. ANOVAs were all performed in RStudio using the function "anova_test" from the rstatix package. Model training and evaluation was completed in Python using the Scikit-learn package. All UMAPs were generated in python with the function umap from the Scikit-learn package. Data for stationary behavior calls is excluded from figures for clarity, however available in the supplemental materials.

### Pharmacology

Neural activity was altered using the established potassium channel blocker; 4-aminopyridine (4-AP; Sigma 275875) and the GABA receptor agonist, muscimol (Sigma; M1523). Stocks were prepared in nanopure water and diluted to working concentrations in E3h embryo media. To test behavioral effects, larvae were treated in two scenarios. First, larvae were treated from 2–4 days post fertilization (dpf) with 400μM 4-AP and 200μM muscimol that were replaced daily. At 4 dpf, larvae were removed from drug and put in E3h embryo media until behavior testing. Second, larvae were treated on the day of behavior testing at 6–7 dpf. Larvae were treated with 800μM 4-AP or 400μM muscimol for one hour prior to behavior testing. Individuals remained in the drug for the duration of behavior testing.

## Results and discussion

### High throughput anatomical key point identification of stimulus locked and spontaneous larval zebrafish movements

Our goal was to establish a high throughput pipeline to identify and characterize larval zebrafish motor responses. To approach this task, we started with a proprietary imaging platform with a multi-camera array microscope (MCAM) from Ramona Optics, which allows imaging of larval zebrafish in 24- and 96-well plate formats, which we assessed as high throughput for a motor call incorporating strategy. The imaging specifications of this system also allow relatively high resolution (3072x3072 per camera) and high speed (max 160 fps) imaging and tracking of larval body positions across acquisition frames using 8 key points positioned using a pose estimation algorithm originally based on DeepLabCut [58] (Fig 1A). We reasoned that this positional tracking data could be leveraged to extract motor kinematics and thus provide high throughout characterization of larval zebrafish motor performance via extracting distinct motor responses.

Our first task was to test the veracity of the pose estimation outputs and establish that the built-in positional tracking reliably detected events consistent with motor bouts. We performed recordings of larvae in both 24 well and 96 well plates. In 24 well plates we recorded 72 fish given an acoustic stimulus, 120 fish given a 2 second light off stimulus using



**Fig 1. High throughput tracking of zebrafish kinematics.** A. Representative labeling of 8 key point pose estimation of zebrafish along the body and representative traces for each key point over a 25 msec duration. Color indicates key point identifier. B. Key point tracking model error for both 24-well plate and 96-well plate tracking models. Each bar represents the average distance error for inferred key point locations compared to a manual label. Green dotted line indicates an error of 75 µm, magenta dotted line indicates an error of 125 µm, and the cyan dotted line indicates an error of 175 µm. Color indicates key point location coinciding with the labels in (A). C. Representative tail angle calculation for a recording where a stimulus was applied. Color indicates angle. D. Depiction of acoustic stimulus tracking assay where larvae are recorded for 10 seconds and given at 0.1 second tap stimulus to the 4 corners of the well plate at 5 seconds. Image denotes a representative average projection of a larvae recorded during the onset of the tap stimulus. E. Average of the maximum absolute value of instantaneous heading change for among all individuals recorded in a 24-well plate (N = 72). F. Bar chart representation of average maximum absolute value of heading change for given intervals 200 msec prior to and to the onset of the tap stimulus (purple, N = 72), from the onset of the tap stimulus extending 200 msec (blue, N = 72), and 200 to 400 msec after the onset of the tap stimulus (mint, N = 72). G. Average absolute maximum caudal tail angle across the recording period. H. Same as in (F) for caudal tail angle. I. Average maximum speed for the same intervals as in (F, H). J. Depiction of recordings evoking a visual stimulus. Larvae were recorded for 6 repeats of 10 second recordings where

environmental illumination was extinguished for 2 seconds halfway through the recording period. K-O. Same as in (E-I) for visual stimulus recordings (N = 120). P. Depiction of spontaneous behavior recordings where individuals were recorded for one-minute intervals repeated 10 times for a total of 10 minutes of recording. Each recording was separated by 3 minutes. Q. Representative identification of bouts based on speed using a threshold of 5 mm/s (cyan) R. Representative instantaneous change in heading direction from the bouts classified in (Q). Average S. maximum speed or T. change in heading direction for all bouts classified using a threshold of 2 mm/s (N = 120). U. Proportion of bouts able to be identified using the 2 mm/s threshold from Q-T. Error bars represent mean± SEM. * indicates p < 0.05.

larvae from 3 independent clutches, and 144 fish across 6 clutches recorded for 10 1-minute intervals with no stimulus. For 96 well plate recordings, we recorded 672 fish across 7 clutches for the acoustic stimulus, and 384 fish across 4 clutches given a 2 second light off stimulus. In either the acoustic stimulus or visual stimulus recordings, groups were exposed to 6 repeats of the stimulus with 120 seconds in between recordings to prevent habituation among recordings. First, we measured the accuracy of the automated key point positioning by manually labeling key points in videos of larval zebrafish in 24- and 96-well plate recordings. We found the average error of key point placement was 1.84 pixels (69.59 µm) and 3.52 pixels (133.12 µm) for 24- and 96-well plate larval zebrafish recordings, respectively (Fig 1B). The largest positional error, which is observed at the tip of the tail during 96 well plate recordings, is roughly equivalent to 5% the body length of a 5–7 dpf larval zebrafish, which we measured to be about 3.85 mm in the 96 well plate and 3.56 mm in the 24 well plate acquired from the MCAM software. We noted that for both plate formats larger errors were mainly on the tail key points. We posit this higher tail error is likely due to reflections from the sides of the wells, which would likely be higher in smaller wells where the animal is, by default, spending more time proximal to the walls, and this prediction is consistent with our 96-well plate measures. Based on this accuracy data, we propose the key point positioning is likely viable for classifying kinematic events as distinct larval zebrafish motor responses.

The next task was to determine if key point positional tracking was extracting positional data relevant to spontaneous and stimulus evoked larval motor outputs. The built-in pose estimation algorithm within the Kestrel MCAM system extracts coordinates from each key point which are then used to derive secondary metrics typical for trajectory analysis, such as speed, change in heading direction, or tail angle (Fig 1C). Therefore, we focused on these metrics to establish whether key point tracking was sufficient to capture motor output. Acoustic startle was selected as auditory stimuli and is well established to evoke a rapid and large body angular change in larval zebrafish referred to as a C-start [10,63]. Therefore, we recorded for 10 seconds and delivered a controlled 0.1 second auditory stimulus as a tap to the four corners of the 24 well plate, which elicited a robust motor response (Fig 1D). From this stimulus, a rapid change in heading direction was observed (Fig 1E). We broke down the recording into three intervals: pre-stimulus, stimulus, and post-stimulus representing 200ms windows prior to, at stimulus onset, or proceeding the stimulus, respectively. The average change in heading direction following an acoustic tap stimulus was significantly larger than baseline behavior (Fig 1F, t(142)=15.593, p < 0.0001). From this recording we tracked the caudal key point tail angle and speed, which showed similar rapid changes with stimulus onset consistent with auditory startle responses (Fig 1G-I, caudal tail angle: two-tailed t-test t(142)=13.471, p < 0.0001; speed: two-tailed t-test t(142)=15.214, p < 0.0001). Next, we presented a visual startle by extinguishing environmental light, known to elicit a large angular body change known as an O-bend [13,34,35]. This visual stimulus also elicited robust changes in heading direction, tail angle, and speed with longer latency as previously reported [24] (Fig 1J-O, Heading angle change, t(238)=−17.971, p < 0.0001; Tail angle: t(238) = −18.483, p < 0.0001, Speed: t(238) = −11.249, p < 0.0001). Similarly, we repeated these two tracking assays in the 96 well plate format and extracted consistent trends (S1 Fig). These data demonstrate that the built-in key point pose estimation calculations and output metrics reliably detected kinematics consistent with sensory-evoked motor responses.

Last, we performed a recording series of 60 second intervals repeated 10 times for a total of 10 minutes under constant illumination with no additional stimuli to determine if these same tracking parameters would similarly detect spontaneous motor outputs (Fig 1P). During typical exploratory behavior in a uniform environment, larvae predominately utilize routine turns (R-turns) and scoots used for small angular reorientation and forward movement, respectively [25,64]. We recorded

144 larvae, which we manually characterized bouts of movement that we defined using a 2 mm/s speed threshold that extended for at least 5 frames (31 msec). Using this method, we were able to extract 362,449 unique bouts of movement across all trials for all individuals making up about 27% of all time recorded. Detected bouts and are distinguished by changes in speed and heading direction (Fig 1Q-U). These results establish that the built-in key point tracking outputs used reliably detect spontaneous and evoked behaviors in larval zebrafish high throughput well plate format recordings and therefore will be sufficient for utilizing machine learning classification of unique behavioral types.

## Semi-supervised modeling of path trajectories delineates distinct zebrafish motor outputs

Our next goal was to leverage key point derived kinematic data to develop a model to identify specific motor maneuvers including auditory startle, visual startle, R-turns, and scoots. While these behaviors are not totally inclusive of the nearly 200 distinct behaviors zebrafish engage in, they define a representative subset of bouts that larval zebrafish naturally utilize in their environment without the addition of external manipulations [26,28]. Scoots and R-turns have been utilized in research to investigate the units that make up larger behaviors such as light search strategies, environmental navigation, and spatial orientation, all of which have underlying neural counterparts which are understood by the ability to visualize and measure the behavior [24,26,42,64,65]. Acoustic startle responses and visual startle responses have been utilized to investigate a myriad of genetic and neurodevelopmental disorders, mechanisms underlying pre-pulse inhibition and habituation, or for large scale chemical screenings which influence the functioning of the respective sensory system [40,66–72].
To do this we developed a machine learning model for analyzing windows of key point tracking data and trained a random forest classifier model for labeling windows of key point coordinates with specific behaviors. Our goal was to identify when zebrafish are engaging in stationary behavior, scoots and R-turns, and stimulus evoked behaviors. From our recordings, we identified behaviors resembling stimulus evoked behaviors, such as C-starts and O-bends, occurring without the presence of a stimulus. Therefore, we opted to define these behaviors as "like" behaviors which identifies behavior resembling that of a classic C-start or O-bend. This is intended to capture the similarities of these well-defined behaviors while additionally allowing for classification of non-stimulus evoked behaviors that zebrafish engage in such as high-speed turning and large angular changes when not exposed to a stimulus. To identify motor outputs for classification we organized data into a sliding window of 40 frames and preprocessed coordinate data from these windows through an egocentric alignment to create a larva focused coordinate system applied to all frames in the sampling window (Fig 2A-B). To visualize the underlying structure of this data we utilized the uniform manifold approximation (UMAP) algorithm to reduce dimensionality and visualize in 2 dimensions the transformed data and color coded by window order which would allow us to visualize how sequential windows orient in space. For both 24- and 96-well plate recording data we were able to visualize organization of sequential datapoints suggesting machine learning models would likely be able to differentiate distinct behaviors from our 40-frame data transformation process. We tested a variety of window lengths from 10 to 80 frames long spanning 62.5 msec to 500 msec to find a duration that balanced temporal resolution and significant structuring of data (S2A Fig). Through this we determined that a 40-frame window covering 250 msec of tracking data gave us substantial temporal resolution to extract individual motor events while having enough structure to cluster behaviors.

Initially, we aimed to use unsupervised clustering to extract motor events as others have shown can extract unique movements that zebrafish engage in [26,27]. We utilized K-means clustering to define 9 clusters within both of our well plate configurations, which we reasoned would allow for us to identify the four behaviors of interest as well as stationary behavior and leaves 4 clusters which we would deem outliers of behaviors that don't reasonably represent any of our selected behaviors. However, we failed to readily define any behaviorally relevant patterns to these clusters aside from stationary behaviors (S2 Fig B). As others have demonstrated, unsupervised learning most notably identifies zebrafish movement patterns rather that namable behaviors, which are made of up sequences of movement patterns [26,29]. Therefore, we used these clusters to extract video clips of larvae when they were moving or while they were stationary which facilitated the process of manually classifying behaviors for supervised learning. We manually categorized



Fig 2. Supervised learning identifies and classifies unique types of motor behavior. A. Depiction of egocentric alignment where from every time-point a transformation is applied to the rest of the frames in the window. Initially larvae are rotated so their heading direction is facing north and the center is translated to become the origin image, then all key points are translated to the origin. B. This transformation is then applied to the subsequent 39 frames in the window for a 40-frame window. C. Representative identification of movement bouts detected from unsupervised clustering with K-means clustering (k = 9). D-E. Confusion matrices for both 24-well plate configuration and 96-well plate configuration RFC models. Color indicates proportion of the classification type used for comparison in the confusion matrix. True indices are feature vectors from our manual segmentation of behavior while predicted are model classifications. Values indicate the proportion of behavior clips that were classified correctly compared to manual labelling. F-G. 2D UMAP of a subset of recording datasets with behaviors labeled in the classification of F. 24 well plate models and G. 96-well plate model. Percent responding trials based on the presence of at least one call made to acoustic or visual stimuli in either the 24-well or 96-well plate recordings.

approximately 1800 clips into distinct behavioral patterns that we had high confidence represent larval zebrafish behavior based on a combination of published data describing the behaviors or temporal association with provided stimuli. Stationary behavior refers to instances where larvae remain motionless or drift passively without active bodily movement. Scoots are characterized by linear propulsion [25,64], while R-turns involve larvae staying largely in place but adjusting their

heading direction [42]. Acoustic startle-like behavior (AsLB) resembles C-start responses following exposure to a startling acoustic stimulus [37–39], whereas visual startle-like behavior (VsLB) is consistent with O-bend responses triggered by a startling visual stimulus [34,35]. We trained two separate random forest classifier (RFC) models for each 24-well plate tracking data and 96-well plate tracking data, which we found was necessary due to differences in hardware video acquisition and pixel density for 24-well plate and 96-well plate tracking data. In a 96-well plate configuration, we were unable to reliably differentiate scoots and turns for manual labelling during supervised learning, which we surmise is likely due to larvae predominately moving along the curved wall in the 96-well plate, making turning due to following a curved surface indistinguishable from self-generated R-turns. We trained the 24 well plate model with 66 stationary, 102 scoots, 75 turns, 60 AsLB, and 137 VsLB behavior clips. Whereas the 96 well plate model was trained with 432 stationary, 486 movement, 170 AsLB, and 309 VsLB behavior clips. From our manually labeled video clips, 90% of these were used to train the model while the remaining 10% was used to test the accuracy of classification. We found that our 96-well plate model had an overall accuracy of F1 = 0.985 (Fig 2D) while our 24-well plate model had F1 = 0.979 for all classified behaviors in the test set (Fig 2E). There were no futher attempts to optimize these models following the test series. Reclassifying the original UMAP data based on outputs from our motor call models showed distinct spatial groupings for differently labeled behaviors (Fig 2F-G). This data suggests our models successfully extracted kinematic parameters consistent with distinct motor responses as defined by our supervised inputs.

Upon initial application of our motor call models, we noticed the sliding window model, where data was processed across a 40 frame window in a frame-to-frame manner caused shifts in the position of motor calls and in other instances imposed frame-by-frame alternating behavior calls impossible to represent real biological action. We reasoned these artifacts were due to the 250 msec duration of each window and the noise from raw output where behaviors can happen anywhere within those 250 msec. To mitigate these artifacts and remove noise from the motor behavior calls, we enacted a "majority voting" method to align motor calls to real time and average out non-biological high fluctuation. We completed this "majority vote" approach by aggregating all windows that overlap a specific timepoint and selecting the most frequently appearing motor call at that time point (S2 Fig C). This not only removes noise and spurious motor calls from the model output but aligns motor calls to the time variable and centralizes calls to the timepoints where the zebrafish is most likely engaging in that behavior (S2 Fig D).

We then sought to compare how evoked startle behaviors differ between the well plate formats for the tested stimuli. We completed this by determining how many individuals responded to either the acoustic or visual stimuli between the well-plate formats. We defined responsive behavior as any individual larva having at least one window called for the elicited behavior (AsLB for acoustic stimuli, VsLB for visual stimuli) within the stimulus onset window. For this, we defined larvae as responding to an acoustic stimulus if they had a response within 250 msec following the elicited tap, and for visual stimuli, if they had a response within 1.0 second following the light turning off. These windows exceed the established latency for each behavioral response, so by using these windows we capture all larvae responding to the respective stimulus [11,34,35,67]. We classified responders responses in two ways, 1) a call matching the elicited behavior was made during a single video for a single larva (ex. One AsLB call within 250 msec of acoustic stimuli for one fish in one video), or 2) at least one of the 6 replicate recording had a matching call (ex. For each fish if at least one of six videos had a response). In the 24 well plate format we found that across all trials, larvae responded 43.1% of the time to the acoustic stimulus, which accounted for 87.5% of all larvae (Fig 2H; S2 Fig E-F). Likewise, across all trials of visually evoked stimulation, larvae responded to 77.5% of the time which accounted for 96.7% of the tested population. In our 96 well plate configuration, larvae had responses to acoustic and visual stimuli 29.6% and 30.9% of the time accounting for 64.1% and 61.7% of larvae respectively. Typically, zebrafish respond to acoustic or visual startling stimuli ~70–80% of the time [24,67,68]. While we found consistent results in the 24 well plate for responses to visual stimuli, responses to acoustic stimuli were nearly half as frequent as expected. Although eliciting acoustic startle responses with a mechanical tap isn't as common as other methods such as using a shaker or speaker, others that have employed this method report response

rates of 70–100% [67,73,74]. However, the differences of these other methods aren't exactly comparable as they focused mechanical stimulation for head fixed larvae in individual petri dishes, rather than broad stimulation across a 24 or 96 well plate. Overall, larvae as a population respond to acoustic stimuli with expected rates, yet not consistently among sequential trials, suggesting that the response rate we recorded may partially be due to the well plate format. These results establish the ability for our models to correctly identify evoked larval zebrafish behaviors.

## High throughput identification of larval zebrafish spontaneous and stimulus evoked motor responses

Next, we applied our model to well plate recordings to extract and classify motor behavior utilization during stimulus evoked and spontaneous behavior and quantify kinematic features from different extracted motor calls. To understand how different motor call types are represented, we extracted sequences of motor calls. We defined bouts as sequences of calls of the same identity that exceeded 5 consecutive windows. For recordings of larvae exposed to an acoustic stimulus we only considered bouts that surrounded the onset of the tap from 5.0 to 5.25 seconds with the tap occurring at 5.0 seconds in our recording (Fig 3A). Identified bouts of the same motor call were pooled. We found that bouts containing AsLB have increased maximum changes in heading direction and speed compared to general movement (non-startle-like associated movement) and stationary classifications (Fig 3B-C, Heading: one-way ANOVA: $F_{(1,258)}$=309.936, $p < 0.0001$, Speed: one-way ANOVA: $F_{(1,258)}$=211.627, $p < 0.0001$). The rapid speed of auditory startle behavior we observed is consistent with other reports describing this behavior [11,67]. This trend was consistent across the 96-well pate tracking and behavioral classification model and consistent with our manual validation (S3 Fig A-C; See Fig 1).

Similarly, we characterized bouts in response to visual stimulation. Zebrafish visual startle responses have a longer latency than acoustic responses, so we examined an interval from the onset of environmental light extinction to one second post-stimulus [13,14] (Fig 3D). We found distinct kinematics make up all bouts within this window (Fig 3 E-F: Heading: one-way ANOVA: $F_{(1,427)}$=769.581, $p < 0.0001$; Speed: one-way ANOVA: $F_{(1,427)}$=791.925, $p < 0.0001$). VsLB calls make up the largest change in heading direction as expected following a loss of illumination stimulus consistent with how O-bends present where zebrafish have a large reorientation of their heading direction contorting their body to resemble the letter "O" [24,35]. Interestingly, we found that AsLB calls were made during visual startle evoking stimuli, although at fewer numbers (AsLB = 34 bouts; VsLB = 115 bouts). Unlike comparisons between VsLB and AsLB during acoustic stimulation where heading changes between AsLB and VsLB do not differ, we found that there's a significant increase in heading change from AsLB to VsLB calls suggesting that AsLB calls made for visual startle evoking stimuli are weaker responses or engagement of visual startle behaviors (t(147)=−2.332, $p = 0.0309$). Comparable patterns were observed for both stimuli recordings from using the 96-well plate experiments and manual validation (S3 Fig A-D; See Fig 1).

Lastly, we surveyed spontaneous motor response utilization in a uniform environment. For this, we used the long recordings previously described of 10 repeated sequential 1-minute recordings for a cumulative 10-minute recording of spontaneous behavior (Fig 3G). From these long recordings, across 144 individuals, 58.3% of all recorded behavior represented bouts of movement in the form of scoots or R-turns (Supplemental Fig 3E). We were able to extract a total of 79683 unique bouts of movement comprised of 48789 (35.7%) scoots, 22308 (16.3%) R-turns, 4403 (3.22%) AsLB, and 4183 (3.06%) VsLB (Fig 3H). Individual larvae performed approximately 550 bouts over the 10 minute recording, or 1 bout of movement every second consistent with other findings [75]. On average, all call types exceed 150 msec duration with turns making up the shortest duration compared to all other maneuvers, which we found to be within the ranges observed by others spanning 100–200 msec [32,75] (Fig 3I, 1-way ANOVA $F_{(1,540)}$=4.869, $p = 0.028$). Next, we showed what motor parameters make up these call types by looking at magnitude of heading direction change and speed during the bout. Both the AsLB and VsLB accounted for the largest maximum change of heading direction and speed while non-startle like behaviors maintained weaker changes in heading direction and speed (Fig 3J-K). Collectively, our data shows that our model is effectively classifying larval zebrafish motor behavior from both stimulus evoked and spontaneous larval zebrafish behavior, and that our models' classified motor bouts represent distinct forms of motor behavior.

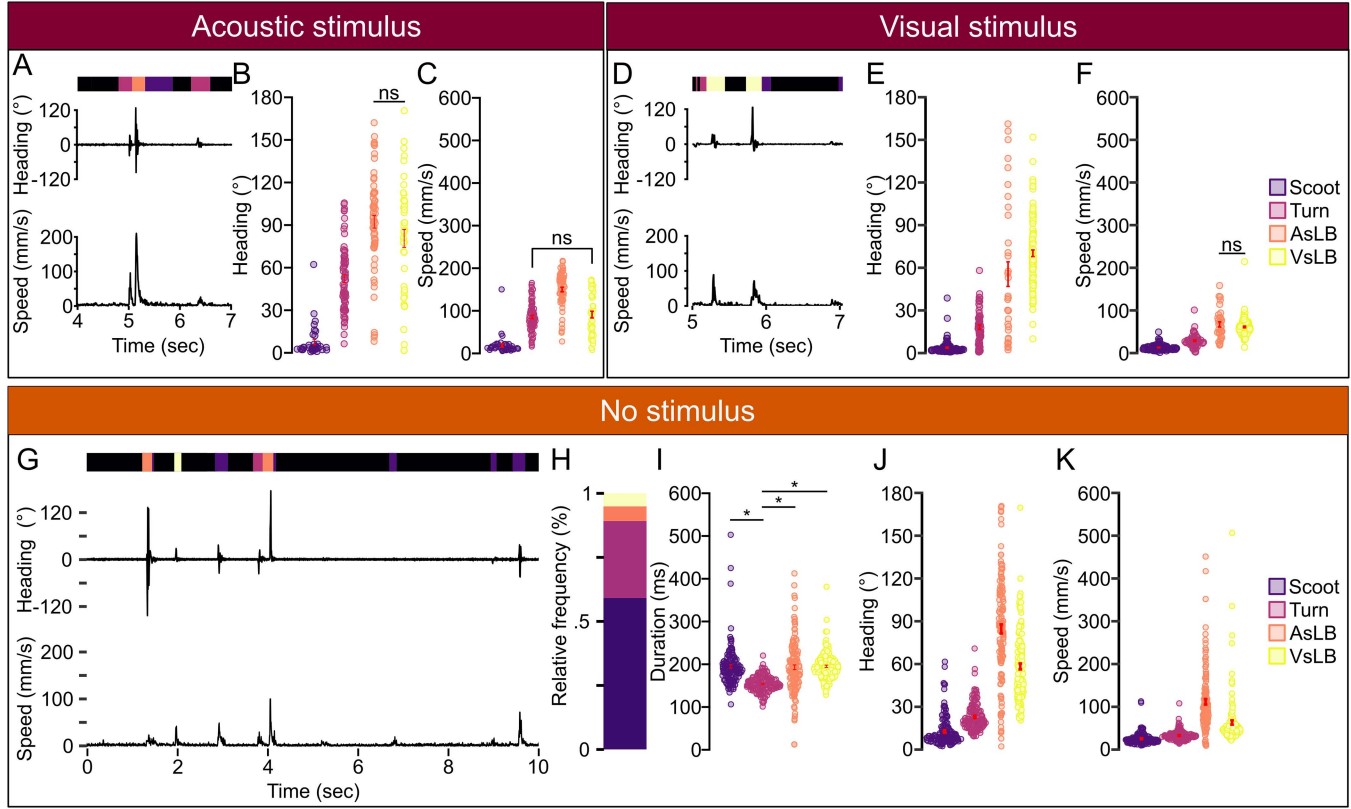

**Fig 3. Classified behaviors show different underlying kinematics.** A. Representative instantaneous heading change and speed for acoustically evoked behavioral recordings in the 24-well plate recording. Bar above heading change indicates the sequence of classified behaviors for each timepoint (black: stationary, dark purple: movement, orange: AsLB, yellow: VsLB). B. Average of the maximum absolute value of heading change across different classified behaviors within the stimulus window for acoustic startle in 24-well plates from 5.0 to 5.25 seconds (Fish N = 72, Behavior: Stationary: N = 49 (19.8%), Scoot: N = 28 (11.3%), Turn: N = 70 (28.3%), AsLB: N = 61 (24.7%), VsLB: N = 42 (17.0%).). C. Average maximum speed across different classified behaviors in 24 well plates. D-E. Same as in (A-C) for visually evoked behaviors in 24 well plates (Fish N = 120, Behavior: Stationary: N = 120 (38.2%), Scoot: N = 103 (32.8%), Turn: N = 57 (18.5%), AsLB, N = 34 (10.8%), VsLB: N = 115 (36.6%). In panels B-C and E-F, percentage in parentheses indicates proportion of all bouts of that type among fish during the stimulus window. Stationary is excluded from the figure for clarity). The window used for E, F was from 5.0 to 6.0 seconds. G. Representative selection of 10 seconds of recording from a total 60 seconds of recording showing instantaneous changes in heading direction and speed. Bar above heading direction indicates the classified behavior for each timepoint (black: stationary, purple: scoot, maroon: turn, orange: AsLB, and yellow: VsLB). H. Relative proportion of classified behaviors excluding stationary behavior (Fish N = 144, Behavior: Stationary: N = 144, Scoot: N = 138, Turn: N = 136, AsLB: N = 130, VsLB: N = 136). I. average bout duration for each classified behavior type. J. Average maximum instantaneous speed across select bout types. K. Average magnitude of maximum instantaneous change in heading direction for select bout types. Stationary is excluded from the figure for clarity. Error bars represent mean ± SEM. * indicates p < 0.05. Comparisons where * is not located means p < 0.05.

## Modulators of brain activity alter bout utilization and not motor kinematics

Next, we wanted to test if motor modulation could be detected by our model to further verify the ability to characterize motor performance. To do this, we used different pharmacologic agents with well-characterized physiological impacts. We selected two systemic modulators of brain activity, 4-Aminopyridine (4-AP) which is a voltage activated potassium channel blocker that is used to induce seizure like activity in other models [22]. In addition, we also used muscimol, which is a GABA$_A$ receptor agonist which increases inhibition in the brain and has been shown to dampen behavioral output in zebrafish [22]. In our previous work, we have also used both of these agents and identified doses that effectively modulate neural activity without causing gross developmental or behavioral deficits [22], which are doses used in the current study

for all treatments. The goal was to ensure larvae used typical motor responses whereas higher concentrations of neuro-modulators could induce atypical motor behavior and likely abnormal bouts [76–78].

We performed a paradigm where larvae were treated with the respective drug for an hour before and throughout the duration of the recording, or controls which are placed in E3h embryo media for the duration of recording (Fig 4A). We then exposed larvae to 6 repetitions of either an acoustic of visual stimulus in the form of a tap for 0.1 seconds or removing environmental illumination for 2 seconds identical to our previous stimulus evoking assays. For both stimulus types we found changes in the frequency of motor response utilization. Compared to controls, muscimol treated larvae had more responses to acoustic stimuli, while 4-AP treated larvae were less responsive to both acoustic and visual stimuli (Fig 4B; S4 Fig A-B). We repeated this experiment in the 96 well plate configuration for comparison; however, we found it did not capture changes in responsiveness among treatment types, likely due to the natively lower responsiveness in 96 well plates (S4 Fig C-H; See S2 Fig). Our goal was to understand how these drugs can change the manner through

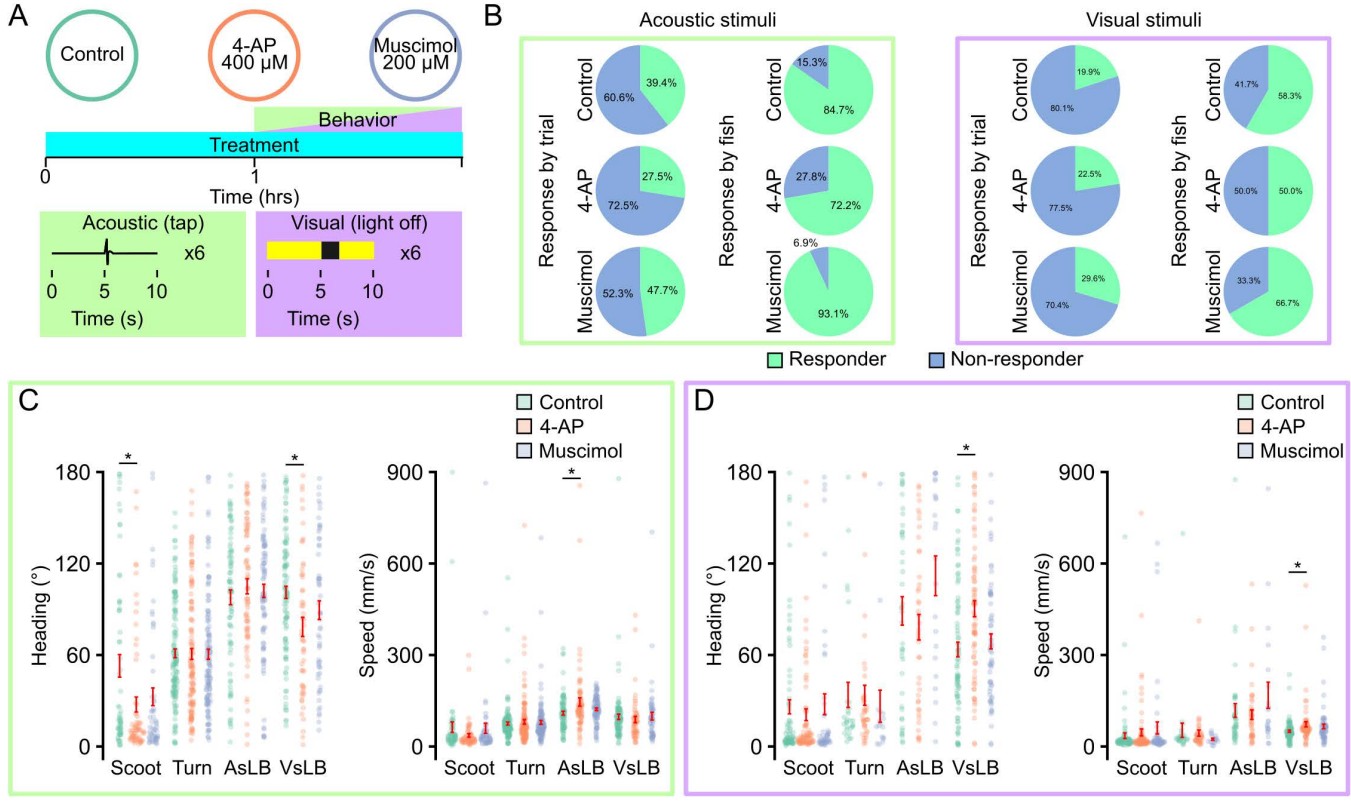

**Fig 4. Neural activity modulators alter underlying behavioral kinematics.** A. Depiction of drug treatment paradigm. Larvae are treated for an hour prior to behavior testing where they are recorded for 6 repeat acquisitions while given either an acoustic tap stimulus or a visual light off for 2 seconds stimulus. Larvae remain in their respective pharmacologic for the duration of behavior testing. Dosages for each drug treatment and response rates across all recording trials for each drug in either an acoustic stimulus recording (green outline) or a visual stimulus recording (purple outline). B. Recordings are classified as responsive if they contain at least one call from their respective stimulus window (Acoustic = 5.00–5.25 seconds; Visual = 5.0-6.0 seconds). Light green indicated proportion of responsive trials while light blue indicates proportion of non-responsive trials in a 24 well plate. C-D. Average of the maximum absolute value of heading change or maximum instantaneous speed across behavior classifications and drug treatments for C. acoustic evoked behavior (green outline, Control Fish N = 144, Behavior: Scoot: N = 65, Turn: N = 123, AsLB: N = 75, VsLB: N = 54; Fish N = 144, Behavior: 4-AP: Scoot: N = 61, Turn: N = 137, AsLB: N = 83, VsLB: N = 105; Fish N = 144, Behavior: Muscimol: Scoot: N = 57, Turn: N = 129, AsLB: N = 93, VsLB: N = 58) or D. visually evoked behavior (purple outline, Control Fish N = 144, Behavior: Scoot: N = 82, Turn: N = 29, AsLB: N = 42, VsLB: N = 83; 4-AP Fish N = 144, Behavior: Scoot: N = 82, Turn: N = 40, AsLB: N = 42, VsLB: N = 83; Muscimol Fish N = 144, Behavior: Scoot: N = 52, Turn: N = 17, AsLB: N = 22, VsLB: N = 65). Error bars represent mean ± SEM. * indicates p < 0 05.

which zebrafish engage in different behaviors. For each recording, zebrafish were exposed to the stimuli, either acoustic or visual, on six repeated trials. However, zebrafish can engage a different behavior over subsequent trials. To overcome this, we extracted specific motor responses, either scoot, turn, AsLB, or VsLB from fish and compared them across drug treatment groups. By extracting and grouping distinct motor calls, we found no difference in heading direction within the stimulation period for acoustically evoked behavior among drug treatments (two-way ANOVA F(2,1375)=0.088, p=0.915). For visually evoked responses, we observed consistent results across treatment groups and motor call types, with no significant changes in the magnitude of maximum speed among larvae (two-way ANOVA F(2,1056)=1.56, p=0.211). However, significant changes to maximum heading direction change was observed among drug treatments and call types (two-way ANOVA: F(2,1056)=7.076, p=0.0009). The 4-AP treated larvae showed significantly larger maximum changes in heading direction following engaging in a VsLB motor bout compared to controls suggesting that 4-AP treatment influences the way zebrafish engage in visual startle behavior (t(155)=3.774, p=0.0007).

Last, we wanted to confirm our models would detect changes in spontaneous motor bout utilization following exposure to neuroactive drug exposure. We came up with two paradigms for investigating this. One where we pretreat larvae only during development or larvae that are immediately pretreated and tested in drug. We maintained the same drugs and concentrations as previously used. For the first paradigm of pretreating larvae only during early development, we treated individuals with their respective drug from 2–4 days post fertilization (dpf) and tested behavior at 7 dpf. We chose this duration for treatment as we have previously shown a critical period exists during this period where motor behavior can be altered by the environment [22]. Our model detected a cumulative of 109,943 unique motor calls across all treatment groups. On average, active motor calls accounted for 58.93% of all recording time (Control=56.9%, 4-AP=61.0%, Muscimol=58.9%; S5 Fig A). For each treatment group we did notice that drug treated larvae experienced more startle like activity compared to controls, whereas controls had more scoot and R-turn movements (Fig 5B). Of all motor call types, drug treated larvae had similar durations of bouts except for 4-AP treated larvae that exhibited longer duration scoots compared to control (t(142)=5.265, p<0.0001) and muscimol (t(141)=5.055, p<0.0001) treated larvae (two-way ANOVA F(2,841) = 4.939, p=0.007) (Fig 5C). We then broke down these motor calls into underlying kinematic metrics such as magnitude of heading direction change and speed. We found that 4-AP treated larvae exhibited larger maximum changes in heading direction for scoots, turns, and VsLB, as well as larger maximum speed during bouts for the same motor classifications (Fig 5D-E) (Heading: two-way ANOVA: F(2,1057)=12.778, p<0.0001; Speed two-way ANOVA: F(2,1057)=8.434, p=0.0002). Muscimol treatment only resulted in larger changes in heading direction and speed for VsLB classified bouts.

For larvae recorded in drug we were able to extract a total of 109198 unique motor calls (Control=33,837, 4-AP=40,426, Muscimol=34,935). Overall, this made up 60.3% of all recording time (Control=58.3%, 4-AP=62.8%, Muscimol=59.5%) (Supplemental Fig 5B). These motor calls were analyzed as above. Recording spontaneous behavior in 4-AP we observed larvae exhibited more startle-like bouts (AsLB, VsLB) compared to controls and muscimol treated larvae (Fig 5G). 4-AP treated larvae generally exhibited longer bout durations, greater changes in heading direction, and increased speed across behaviors, while muscimol treated larvae either had no change compared to controls or reduced heading changes and speed across bout types, with both treatments significantly altering locomotor metrics (Fig 5H–J). We then looked at maximum speed traveled within each bout among fish finding that 4-AP treated larvae were faster for all bouts except turns and AsLB while muscimol treated larvae were slower for all bouts except turns. These data highlight the biologic applications of our trained machine learning models being able to identify changes in motor performance and responsiveness from systemic neural activity modulators during both stimulus-evoked and spontaneous behaviors.

## Conclusions

Here we developed two machine learning models for characterizing an array of behaviors (stationary, scoot, R-turns, AsLB, VsLB) in a high throughput system. Our aim was to bridge the gap between rudimentary, high throughput behavioral analyses and high throughput, but extremely complex analyses. We developed a protocol that builds upon recent

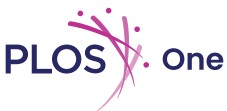

Fig 5. **Neural activity modulators alter underlying kinematics of classified spontaneous behaviors.** A. Depiction of early drug treatments. Larvae are treated with either 4-AP, muscimol, or nothing from 2-4 dpf. They are then removed from the drug and behavior testing took place at 7 dpf. Larvae are tracked for 10 repeats of 1-minute recordings separated by 3 minutes. B. Relative proportion of classified behavior types excluding stationary classifications for control, 4-AP, and muscimol treated larvae. Color indicates behavioral classification (purple: scoot, maroon: turn, orange: AsLB, and yellow: VsLB). C-E. Bar graphs measuring average C. bout duration, D. average maximum absolute value of instantaneous heading direction, or E. maximum speed across select behavioral classifications and drugs (Green: control, Scoot: N=72, Turn: N=72, AsLB: N=69, VsLB: N=72, Orange: 4-AP, Scoot: N=72, Turn: N=72, AsLB: N=67, VsLB: N=71, Muscimol: blue, Scoot: N=71, Turn: N=71, AsLB: N=69, VsLB: N=69). F. Depiction of same day drug treatments. Larvae are treated with either 4-AP, muscimol, or nothing for an hour prior to behavior testing, then they remain in the drug for the duration of



behavior testing. Larvae are tracked for 10 repeats of 1-minute recordings separated by 3 minutes. G-J Same as in B-E for larvae behavior tested while being treated (Control: Scoot: N = 70, Turn: N = 71, AsLB: N = 69, VsLB: N = 69; 4-AP: Scoot: N = 72, Turn: N = 72, AsLB: N = 71, VsLB: N = 71; Muscimol: Scoot: N = 72, Turn: N = 71, AsLB: N = 65, VsLB: N = 69).

engineering advancements through the utilization of the Ramona Optics Kestrel which provides high-throughput and high-resolution imaging of zebrafish behavior and the wide availability of machine learning frameworks which allow for custom trained models that can be applied to behavioral data and workflows. We designed our models to further leverage zebrafish as a model for understanding neural mechanisms underlying behavioral regulation and sensorimotor responsiveness.

Others have developed similar models for characterizing zebrafish behavior, yet with a focus on unique, context specific, behavioral phenotypes like seizure behavior, or different movement patterns associated with wandering swims or large angle turns [26,27,29,79]. Our work was aimed at general zebrafish behavior; hence we chose to only classify representative forms of spatial navigation (Scoots and R-turns), and commonly used stimulus evoked behaviors (AsLB, VsLB). While a variety of machine learning applications have been developed for investigating zebrafish behavior, the ones developed here significantly differ. One such model was designed to categorize seizure like activity in zebrafish has been developed for the same MCAM platform. In this method the authors used an RFC model to identify specific seizure like movements with chemical treatment or genetic variants [79]. Our model offers comparable outputs, yet with the distinct difference of identifying and classifying typical navigation and responsive zebrafish behavior. For instance, if we were to apply our model to seizure-like behavior responses, we could speculate that no well-defined seizure associated responses would be delineated and potentially erratic startle-like behavior would be observed in the form of either AsLB or VsLB as the most similar behavior based on the training of our models. Such examples highlight the efficacy of machine learning applications for classifying diverse and nuanced patterns.

We designed two supervised models for 24 and 96 well plates. For our 24 well plate model, we included navigational types of movements (e.g., scoots and R-turns). However, in our 96 well plate model we combined these together to form a general "movement" classification of behavior. We note that the merger of scoots and turns in our 96-well plate model limits the utility for studying broader navigational movement yet is ideal for higher throughput stimulus evoked startle type behavior. Secondly, prior work dissuades the use of 96 well plates for longer behavioral recordings which makes the necessity for navigational types of behavior in 96 well plates obsolete. [80] In the current study, we did not test for performance differences among well plates. Others have focused on this topic suggesting that varying well plate diameters can alter behavioral performance. [80–84] Mainly, larger arenas, as is such in 24 well plates, can increase the number of swimming events and potentially influence larvae acceleration, speed, and bout kinematics. [80,81,84] Nonetheless, our models provide diverse utility for assays designed for 24 or 96 well plates.

We then tested the biologic applicability of our models against two known neuromodulatory drugs. Our models revealed variation among different behavioral classifications between 4-AP, which is known to cause hyperactivity, and muscimol, which decreases behavioral output. To our knowledge, no other studies have been completed focused on understanding the bout kinematics following application of these drugs. We reported 4-AP treated individuals exhibiting less responsiveness to startling stimuli. Our method of bath application exposed zebrafish to 4-AP for an hour prior to behavior testing. 4-AP's mechanism of action is to selectively block voltage-gated potassium channels which prolongs synaptic transmission between neurons and the neuromuscular junction. [85] Therefore, is it plausible that this extended exposure is causing fatigue in the zebrafish musculature through prolonged depolarization inhibiting movement, although we did not test this hypothesis. [85] In addition, muscimol treated larvae showed no differences in the way that treated individuals utilize motor bouts during startling stimuli. Typically, muscimol treatment reduces behavioral activity, however when larvae do engage in that behavior, the kinematics and movements resemble untreated larvae. Overall, our results demonstrate that our models can detect changes in motor performance from biologically relevant assays.



The models developed here provide a powerful resource for categorizing zebrafish behaviors, yet some limitations exist. For example, scenarios modulating behavioral performance such as drug treatments, genetic mutants with diminished motor behavior, or states of increased motor output our current model might not completely capture the entirety of motor performance [46,49,86,87]. For example, at high dosages, 4-AP treated larvae can exhibit sporadic behavior [88,89]. In zebrafish this is denoted by erratic movement and large, rapid bending of the body [88,90]. For our models, classified behaviors are categorized among scoot, turns, AsLB, or VsLB and cannot differentiate subtle variations that would distinguish a visual startle from a rapid contortion associated with an acute seizure response. In addition, behaviors like short and long latency startle, which are controlled by distinct neural mechanisms yet are kinematically similar, are difficult to resolve due to the current capture rate [67,91]. Using a sliding window minimizes the chance of missing a response, yet does impose some timing error surrounding identified bouts. However, by implementing our method of majority voting reduces this error to less than 250 msec by identifying the most probable behavior being engaged in at a specific point in time. Last, implementing the current models is currently limited to the Ramona Optics proprietary MCAM and software. Nevertheless, this combination of machine learning models and hardware provide a validated platform for high throughput characterization of zebrafish behavior and provide a broadly applicable template to adapt machine learning models to characterize diverse behaviors for basic and biomedical studies.

## Supporting information

**S1 Fig.   A-E.** Metrics plotted from individuals given a tap stimulus at 5.0 seconds in 96 well plate recordings for **A.** Average maximum absolute instantaneous heading change (N = 643). **B.** Interval averaged maximum change in absolute heading direction **C.** Average maximum change in caudal tail angle **D.** interval averaged maximum change in caudal tail angle, or **E.** interval averaged maximum change in instantaneous speed. **F-J.** Same as in **A-E.** for individuals given a visual light off stimulus for 2 seconds (N = 766). Color indicates interval (purple: 200 msec prior to onset of stimulus, blue: 200 msec immediately after onset of stimulus, green: 200 msec following the stimulation interval). * Indicates p < 0.05.
(TIFF)

**S2 Fig.   A.** UMAP representation of various window lengths. Color indicates the sequential order of window. **B.** Representative unsupervised k-means clusters (k = 9, blue) overlaid on heading angle change (top) and instantaneous speed (bottom) for one fish. **C.** Schematic diagram of window aggregation method where at each timepoint all windows covering that timepoint are aggregated together and whichever window exhibits the majority call becomes the behavioral call for that timepoint. **D.** Comparison of raw output from model labeleing of a representative fish in a 96 well plate (orange) versus majority voting output (magents) compared to manual labeling of behaviors (teal). **E.** Responsive fish based on the presence of a call during the stimulus period across 6 acoustic startle acquisitions in 96 well plate recordings for acoustic and visual stimulation. **F.** Same as in **E.** for a VsLB call during the stimulus period for individuals recorded in the 24 well plate configuration.
(TIFF)

**S3 Fig.   A.** Representative heaving angle change (top) and speed (bottom) aligned with behavioral calls (colored bar) for a 96 well plate individual during acoustic startle. Color indicates behavioral call type **B.** Average maximum heading angle change (left) or speed (right) per behavioral call type (Fish N = 643, Behavior: Stationary: N = 569 (42.2%), Movement: N = 319 (23.7%), AsLB: N = 412 (30.6%), VsLB: N = 47 (3.4%). Number in parenthesis represents proportion of the motor call among all bouts per fish) **C.** Same as in **B** for maximum instantaneous speed. **D-F.** Same as in **A-C** for visual startle (Fish N = 766, Behavior: Stationary: N = 384 (43.0%), Movement N = 252 (28.2%), AsLB: N = 30 (3.4%), VsLB: N = 227 (25.4%). Number in parenthesis represents proportion of the motor call among all bouts per fish). **G.** Relative frequency of call type among larvae including stationary calls. ns indicates p > 0.05.
(TIFF)

**S4 Fig. Relative proportion of behavioral call types excluding stationary calls for A.** acoustically evoked startle behavior or **B.** visually evoked startle behavior among drug treatments during the stimulation window. Proportion of responsive fish or responsive trials indicating one behavioral call for AsLB for **C.** acoustic startle (green outline) or **D.** VsLB for visual startle (purple outline) across 6 recordings for each drug treatment. Same as in **A-B** for 96-well plates during either **E.** acoustically evoked startle behavior or **F.** visually evoked startle behavior. **G.** Average maximum change in absolute heading direction or speed for each call type during acoustic stimulation (Control Fish N = 576: Movement: N = 298, AsLB: N = 209, VsLB: N = 32; 4-AP Fish N = 576: Movement: N = 246, AsLB: N = 186, VsLB: N = 18; Muscimol Fish N = 576: Movement: N = 286, AsLB: N = 198, VsLB: N = 30). **H.** Same as in **G.** for visual startle (Control Fish N = 576: Movement: N = 354, AsLB: N = 53, VsLB: N = 164; 4-AP: Movement Fish N = 576: N = 204, AsLB: N = 50, VsLB: N = 149; Muscimol Fish N = 576: Movement: N = 328, AsLB: N = 46, VsLB: N = 111).
(TIFF)

**S5 Fig. A.** Relative frequency of behavioral call type among drug treated larvae from 2–4 dpf including stationary calls. **B.** Relative frequency of behavioral call type among drug treated larvae tested in drug including stationary calls. Color indicates behavioral call type.
(TIFF)

## Acknowledgments

We would like to thank the whole team at Ramona Optics, Inc. for their assistance with completing this research.

## Author contributions

**Conceptualization:** John Hageter, John Efromson, Eric J. Horstick.

**Data curation:** John Hageter, John Efromson.

**Formal analysis:** John Hageter, John Efromson.

**Funding acquisition:** Mark Harfouche, Eric J. Horstick.

**Investigation:** John Hageter, Brooke Alban, Audrey DelGaudio, Veton Saliu, Monica Wassef.

**Methodology:** John Hageter.

**Supervision:** Mark Harfouche, Eric J. Horstick.

**Writing – original draft:** John Hageter, John Efromson, Eric J. Horstick.

**Writing – review & editing:** John Hageter, Eric J. Horstick.

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
