## [Decision Letter · Decision Letter 0]

23 Jul 2025

PONE-D-25-34999High throughput machine learning pipeline to characterize larval zebrafish motor behaviorPLOS ONE

Dear Dr. Horstick,

Thank you for submitting your manuscript to PLOS ONE.  Peer review found the work to be rigorous and important, but identified a number of concerns that must be addressed prior to publication.  We  invite you to submit a revised version of the manuscript that thoroughly addresses all the points raised by the reviewers.

We look forward to receiving your revised manuscript.

Kind regards,

Yevgenya Grinblat, Ph. D.

Academic Editor

PLOS ONE

Journal Requirements:

2. To comply with PLOS One submissions requirements, in your Methods section, please provide additional information regarding the experiments involving animals and ensure you have included details on (1) methods of sacrifice, (2) methods of anesthesia and/or analgesia, and (3) efforts to alleviate suffering.

3. Please note that PLOS One has specific guidelines on code sharing for submissions in which author-generated code underpins the findings in the manuscript. In these cases, we expect all author-generated code to be made available without restrictions upon publication of the work. Please review our guidelines at https://journals.plos.org/plosone/s/materials-and-software-sharing#loc-sharing-code and ensure that your code is shared in a way that follows best practice and facilitates reproducibility and reuse.

5. Thank you for stating in your Funding Statement:

“This work was supported by National Science Foundation cooperative agreement OIA2242771, National Eye Institute R15EY036226, National Institute of General Medicine P20GM144230 awarded to Eric Horstick and the Office Of The Director, National Institutes Of Health under Award Number R44OD036187; the National Institute of Mental Health under Award Number R43MH133521; the National Institute of Environmental Health Sciences under Award Number R43ES036389; and the National Cancer Institute under Award Number R44CA285197 awarded to Ramona Optics. The content is solely the responsibility of the authors and does not necessarily represent the official views of the National Institutes of Health. Computational resources were provided by the WVU Research Computing Thorny Flat High Performance Computing cluster, which is funded in part by NSF OAC-1726534.”

6. Thank you for stating the following financial disclosure:

“This work was supported by National Science Foundation cooperative agreement OIA2242771, National Eye Institute R15EY036226, National Institute of General Medicine P20GM144230 awarded to Eric Horstick and the Office Of The Director, National Institutes Of Health under Award Number R44OD036187; the National Institute of Mental Health under Award Number R43MH133521; the National Institute of Environmental Health Sciences under Award Number R43ES036389; and the National Cancer Institute under Award Number R44CA285197 awarded to Ramona Optics. The content is solely the responsibility of the authors and does not necessarily represent the official views of the National Institutes of Health. Computational resources were provided by the WVU Research Computing Thorny Flat High Performance Computing cluster, which is funded in part by NSF OAC-1726534.”

7. Thank you for stating the following in the Acknowledgments Section of your manuscript:

“We would like to thank the whole team at Ramona Optics, Inc. for their assistance with completing this research. This work was supported by National Science Foundation cooperative agreement OIA2242771, National Eye Institute R15EY036226, National Institute of General Medicine P20GM144230 awarded to Eric Horstick and the Office Of The Director, National Institutes Of Health under Award Number R44OD036187; the National Institute of Mental Health under Award Number R43MH133521; the National Institute of Environmental Health Sciences under Award Number R43ES036389; and the National Cancer Institute under Award Number R44CA285197 awarded to Ramona Optics. The content is solely the responsibility of the authors and does not necessarily represent the official views of the National Institutes of Health. Computational resources were provided by the WVU Research Computing Thorny Flat High Performance Computing cluster, which is funded in part by NSF OAC-1726534.”

“This work was supported by National Science Foundation cooperative agreement OIA2242771, National Eye Institute R15EY036226, National Institute of General Medicine P20GM144230 awarded to Eric Horstick and the Office Of The Director, National Institutes Of Health under Award Number R44OD036187; the National Institute of Mental Health under Award Number R43MH133521; the National Institute of Environmental Health Sciences under Award Number R43ES036389; and the National Cancer Institute under Award Number R44CA285197 awarded to Ramona Optics. The content is solely the responsibility of the authors and does not necessarily represent the official views of the National Institutes of Health. Computational resources were provided by the WVU Research Computing Thorny Flat High Performance Computing cluster, which is funded in part by NSF OAC-1726534.”

8. Thank you for stating the following in the Competing Interests/Financial Disclosure section:

“The authors declare the following financial and personal relationships that may be considered as potential competing interests: J.E., V.S., M.W, and M.H. have a financial interest in Ramona Optics, Inc.”

We note that you received funding from a commercial source: Ramona Optics, Inc.

9. In the online submission form, you indicated that “All experimental data is retained locally and will be made available upon request.”

10.  Please include captions for your Supporting Information files at the end of your manuscript, and update any in-text citations to match accordingly. Please see our Supporting Information guidelines for more information: http://journals.plos.org/plosone/s/supporting-information. "

Reviewers' comments:

Reviewer's Responses to Questions

**Comments to the Author**

1. Is the manuscript technically sound, and do the data support the conclusions?

Reviewer #1: Yes

Reviewer #2: Yes

2. Has the statistical analysis been performed appropriately and rigorously? 

Reviewer #1: Yes

Reviewer #2: Yes

3. Have the authors made all data underlying the findings in their manuscript fully available?

Reviewer #1: Yes

Reviewer #2: Yes

4. Is the manuscript presented in an intelligible fashion and written in standard English?

Reviewer #1: Yes

Reviewer #2: Yes

5. Review Comments to the Author

Reviewer #1: The manuscript presents a robust, technically sophisticated, and well-validated machine learning-based framework for the high-throughput classification of larval zebrafish motor behaviors. This framework leverages pose estimation data obtained from a multi-camera imaging platform, allowing for the precise quantification of a wide range of motor outputs. The authors successfully address a significant technical bottleneck in the field of behavioral neuroscience, where the classification of complex, dynamic larval behaviors has remained challenging due to technical limitations in both imaging and computational analysis. The proposed methodology is particularly noteworthy for its combination of state-of-the-art computational approaches including semi-supervised clustering and supervised machine learning with a scalable, high-throughput imaging platform. This integration enables comprehensive, objective, and automated classification of both spontaneous and stimulus-evoked motor behaviors in freely swimming zebrafish larvae. Importantly, the authors rigorously validate their behavioral classification approach using well-established pharmacological agents known to modulate neural circuits, demonstrating the biological relevance and reproducibility of their results. Furthermore, the authors provide open-source access to their codebase and detailed methodological descriptions, ensuring that their approach can be readily adopted and further developed by the broader research community. This aspect significantly enhances the impact and translational potential of the work, contributing not only a methodological advance but also a valuable resource for the field. The manuscript is overall of very high quality. The experimental design is sound, the data are comprehensive and clearly presented, and the interpretations are well supported by the evidence provided. The clarity of the figures and the logical flow of the manuscript make it accessible to both specialists in zebrafish neuroscience and a broader audience interested in quantitative behavioral analysis. This reviewer recommends acceptance of this manuscript after the authors addressed the following minor issues, primarily concerning language clarity and additional methodological clarification.

1. Several sections would benefit from editorial refinement to improve clarity and scientific rigor. While the current text is generally comprehensible, it occasionally suffers from unclear sentences.

Introduction, Lines 80: However, these metrics alone do not inform what specific motor outputs are being used.

Suggestion: Clarify what is meant by "inform"; perhaps rephrase as: However, these metrics alone do not allow for the determination of the specific motor outputs utilized by the larvae.

2. Introduction, Lines 83: We show that with this training our model reliably detects these four predominant motor outputs as well as non-moving stationary behavior.

Suggestion: Our training enabled the model to reliably detect these four predominant motor outputs, in addition to stationary (non-movement) behavior.

3. Results, Lines 204: In a 96-well plate configuration, we found that our supervised model could not reliably differentiate scoots and turns, which we surmise is likely due to insufficient roaming space to resolve short forward scoots and reorienting R-turns. Therefore, we combined these classifications into a ‘movement’ class...

This explanation is plausible but would benefit from additional justification. Please specifically followings.

- Please clarify whether this limitation was quantitatively assessed (e.g., by analyzing bout trajectory lengths or angular changes).

- Indicate whether any attempts were made to optimize the classifier to resolve these behaviors despite the spatial constraints.

- Discuss briefly how this class merging may affect the biological interpretation of the movement data in the 96-well format.

4. Throughout the Results (e.g., Lines 197), the terms “acoustic startle-like behavior (AsLB)” and “visual startle-like behavior (VsLB)” are used. While the manuscript later explains that these terms refer to behaviors resembling known stimulus-evoked responses, a clearer introduction to these terms earlier in the Introduction would be helpful.

5. Results, Lines 106: We performed recordings of 728 larvae given a tap stimulus, 504 larvae given a 2-second light off stimulus, and 144 larvae recorded for spontaneous motor output...

Clarification is needed regarding the biological replicates. Please explicitly state whether these numbers refer to the total number of larvae pooled from multiple independent experimental batches, or whether all recordings were performed in a single experimental batch.

Reviewer #2: In Hageter et al, the authors establish new machine learning tools to identify and classify evoked and spontaneous larval zebrafish movements captured using the Ramona Kestrel multi camera array microscope (MCAM). These tools will be of broad use to the growing base of MCAM users in the zebrafish community. The authors provide a solid analysis of and validation of their behavioral classification tools, but several areas should be strengthened and/or clarified prior to publication:

Major areas to address:

• Throughout the manuscript, showing individual data points is strongly preferred. Minimally, violin plots should be used to show distributions of the data. This is particularly important for evoked acoustic and visual responses, which are well-established to have multiple response flavors, with different underlying circuits, and different response kinematics. These responses also are well known to have strong variability between individuals, so the authors would do well to show how their data align with what is known here.

• The authors show many examples in which behaviors differ when fish are testing in 24 versus 96 well plates. Further discussion of these differences is warranted. Why do fish perform slower movements in 96 well plates? Why do their startle responses have shallower angles? Why do UMAPs of behavior groupings differ so much between 24 and 96 well formats (Figure 2F vs G)? A discussion of the pros and cons of each plate format for analyzing different types of behaviors based on the authors’ data would be very informative for the community, as the design of the MCAM and other zebrafish-focused testing platforms relies on multi-well plates to achieve medium-to-high throughput for screening drugs, chemicals, and genes. Putting the current findings in the context of other studies that have reported similar differences in behavior depending on the testing arena size is warranted.

• The authors should show the distributions of response types in Figure 2A and 2B. How many “VsLBs” were called following acoustic stimuli and vice versa? Latency distributions would also be helpful here, as the known distinctions in latency between acoustic and visual startle responses might help to correctly reclassify some “visual” responses to acoustic stimuli and vice versa. The authors replicate this distinction in Figure 1E vs K, so this should be feasible. The crossover in response types between AsLBs and VsLBs, despite the “like” in the acronym, makes these labels somewhat misleading. Perhaps using labels that classify them based more turn angle, speed, and/or latency would be more appropriate.

• Figure 4C,D: it seems that the n’s are defined as responses, but perhaps the more relevant biological unit is the fish. Are some of these differences in heading and speed driven by individual larvae rather than shared changes happening across the population that was tested? This question cannot be answered the way the data is presented.

• Do the drug treatments in Figure 4 change the distribution of response types? It would help to show something similar to Figure 5B.

• The authors’ rationale for the 4-AP and muscimol experiments is to confirm that the MCAM and their deep learning model can detect the behavioral changes induced by these drugs, either acutely or developmentally. Yet the authors do not address whether their findings are consistent with their prior findings and the broader literature on the effects of these drugs.

Minor concerns:

• Would be helpful to specify in the text that Figure 3 uses 24 well plates. The authors could also denote this on the figure itself.

• Are distributions in Fig 5B and G significantly different?

• Some info about how the acoustic stimulus is delivered is needed in the methods section, ideally with some measurement of intensity in decibels or acceleration

Typographical corrections:

• Line 61: Throughput not “throughout”

• Supp Fig. 1E y-axis labels out of order

• Line 150: extraneous “and”

• Line 344: “treated” not “treatmented”

• Line 411: “TL” strain is “Tupfel long-fin” not “Tubingen long-fin”

6. PLOS authors have the option to publish the peer review history of their article (what does this mean? ). If published, this will include your full peer review and any attached files.

**Do you want your identity to be public for this peer review?** For information about this choice, including consent withdrawal, please see our Privacy Policy .

Reviewer #1: No

Reviewer #2: No

---

## [Author Response · Author response to Decision Letter 1]

25 Sep 2025

We thank both reviewers for taking the time to evaluate our manuscript and providing thorough and helpful commentary to improve the work. We appreciate the supportive assessments of the manuscript and the overall enthusiasm for the resources developed. Below we provide a point-by-point response to reviewer requests. Our comments are provided in BOLD text for each reviewer’s comment. Please note any line references made refer to the tracked manuscript version. We also want to bring to the reviewer’s attention the manuscript as been reformatted to Plos1 standards as editorially requested.

Reviewer #1

1. Several sections would benefit from editorial refinement to improve clarity and scientific rigor. While the current text is generally comprehensible, it occasionally suffers from unclear sentences.

Introduction, Lines 80: However, these metrics alone do not inform what specific motor outputs are being used.

Suggestion: Clarify what is meant by "inform"; perhaps rephrase as: However, these metrics alone do not allow for the determination of the specific motor outputs utilized by the larvae.

Thank you for the suggestion and we revised as suggested to -

“However, these metrics alone do not allow for the determination of the specific motor outputs utilized by the larvae.”

2. Introduction, Lines 83: We show that with this training our model reliably detects these four predominant motor outputs as well as non-moving stationary behavior.

Suggestion: Our training enabled the model to reliably detect these four predominant motor outputs, in addition to stationary (non-movement) behavior.

Revised as suggested to “Our training enabled models to reliably detect these four predominant motor outputs, in addition to stationary (non-moving) behavior.”

3. Results, Lines 204: In a 96-well plate configuration, we found that our supervised model could not reliably differentiate scoots and turns, which we surmise is likely due to insufficient roaming space to resolve short forward scoots and reorienting R-turns. Therefore, we combined these classifications into a ‘movement’ class...

This explanation is plausible but would benefit from additional justification. Please specifically followings.

- Please clarify whether this limitation was quantitatively assessed (e.g., by analyzing bout trajectory lengths or angular changes).

- Indicate whether any attempts were made to optimize the classifier to resolve these behaviors despite the spatial constraints.

- Discuss briefly how this class merging may affect the biological interpretation of the movement data in the 96-well format.

Thanks for pointing this out as we needed to make a clarification. For identifying inputs to train our supervised model we could not confidently resolve scoots and turns in the 96 well videos for manual labeling. To better represent this, we changed language in the results to:

“In a 96-well plate configuration, we were unable to reliably differentiate scoots and turns for manual labelling during supervised learning, which we surmise is likely due to larvae predominately moving along the curved wall in the 96-well plate, making turning due to following a curved surface indistinguishable from self-generated R-turns.” Starting at Line 376.

We also included in the conclusions a brief discussion of the implications of this merger:

However, in our 96 well plate model we combined these together to form a general “movement” classification of behavior. We note that the merger of scoots and turns in our 96-well plate model limits the utility for studying broader navigational movement yet is ideal for higher throughput stimulus evoked startle type behavior. Starting at Line 633.

4. Throughout the Results (e.g., Lines 197), the terms “acoustic startle-like behavior (AsLB)” and “visual startle-like behavior (VsLB)” are used. While the manuscript later explains that these terms refer to behaviors resembling known stimulus-evoked responses, a clearer introduction to these terms earlier in the Introduction would be helpful.

Thank you for pointing out this oversight on our part. The introduction to our classified behavior types has been moved to their first usage in text and rephrased for clarity:

“Our goal was to identify when zebrafish are engaging in stationary behavior, scoots and R-turns, and stimulus evoked behaviors. From our recordings, we identified behaviors resembling stimulus evoked behaviors, such as C-starts and O-bends, occurring without the presence of a stimulus. Therefore, we opted to define these behaviors as “like” behaviors which identifies behavior resembling that of a classic C-start or O-bend. This is intended to capture the similarities of these well-defined behaviors while additionally allowing for classification of non-stimulus evoked behaviors that zebrafish engage in such as high speed turning and large angular changes when not exposed to a stimulus.” Starting at Line 321.

As we provided this introduction earlier, we also deleted the original sentence (Line starting at 371) to avoid redundancy.

5. Results, Lines 106: We performed recordings of 728 larvae given a tap stimulus, 504 larvae given a 2-second light off stimulus, and 144 larvae recorded for spontaneous motor output...

Clarification is needed regarding the biological replicates. Please explicitly state whether these numbers refer to the total number of larvae pooled from multiple independent experimental batches, or whether all recordings were performed in a single experimental batch.

Thank you for the suggestion. In the methods we included the clarification that “All experiments are composed of at least 3 independent cohorts of zebrafish.” See lines 105-106.

We also revised the explanation of these groups within the results and conclusions section to:

“We performed recordings of larvae in both 24 well and 96 well plates. In 24 well plates we recorded 72 fish given an acoustic stimulus, 120 fish given a 2 second light off stimulus using larvae from 3 independent clutches, and 144 fish across 6 clutches recorded for 10 1-minute intervals with no stimulus. For 96 well plate recordings, we recorded 672 fish across 7 clutches for the acoustic stimulus, and 384 fish across 4 clutches given a 2 second light off stimulus. In either the acoustic stimulus or visual stimulus recordings, groups were exposed to 6 repeats of the stimulus with 120 seconds in between recordings to prevent habituation among recordings.”

Starting at Line 254.

Reviewer #2

Major areas to address:

• Throughout the manuscript, showing individual data points is strongly preferred. Minimally, violin plots should be used to show distributions of the data. This is particularly important for evoked acoustic and visual responses, which are well-established to have multiple response flavors, with different underlying circuits, and different response kinematics. These responses also are well known to have strong variability between individuals, so the authors would do well to show how their data align with what is known here.

As suggested, we changed all figures to sina plots representing distributions and individual data points. However, this initially created an overrepresentation of density in the stationary data, so we opted to exclude it from visualization. These exclusions are made apparent where applicable. See Fig 3 Legend and lines starting at 458, 465, and Materials and Methods lines 199.

• The authors show many examples in which behaviors differ when fish are testing in 24 versus 96 well plates. Further discussion of these differences is warranted. Why do fish perform slower movements in 96 well plates? Why do their startle responses have shallower angles? Why do UMAPs of behavior groupings differ so much between 24 and 96 well formats (Figure 2F vs G)? A discussion of the pros and cons of each plate format for analyzing different types of behaviors based on the authors’ data would be very informative for the community, as the design of the MCAM and other zebrafish-focused testing platforms relies on multi-well plates to achieve medium-to-high throughput for screening drugs, chemicals, and genes. Putting the current findings in the context of other studies that have reported similar differences in behavior depending on the testing arena size is warranted.

Here we break the comment down into 4 components -

Performance between well plate formats:

Behavioral experiments between 24 and 96 well plates are independent cohorts of zebrafish, therefore our assays and not poised to make direct comparisons among behaviors for different well plate sizes, however our work is not the only example of behavior being altered among different well plate diameters. We added a section to the conclusions to highlight literature that focuses more on the difference of behavior among arena sizes and well plate diameters. Starting at Line 632.

“We designed two supervised models for 24 and 96 well plates. For our 24 well plate model, we included navigational types of movements (e.g. scoots and R-turns). However, in our 96 well plate model we combined these together to form a general “movement” classification of behavior. We note that the merger of scoots and turns in our 96-well plate model limits the utility for studying broader navigational movement yet is ideal for higher throughput stimulus evoked startle type behavior. Secondly, prior work dissuades the use of 96 well plates for longer behavioral recordings which makes the necessity for navigational types of behavior in 96 well plates obsolete. [80] In the current study, we did not test for performance differences among well plates. Others have focused on this topic suggesting that varying well plate diameters can alter behavioral performance. [80–84] Mainly, larger arenas, as is such in 24 well plates, can increase the number of swimming events and potentially influence larvae acceleration, speed, and bout kinematics. [80,81,84] Nonetheless, our models provide diverse utility for assays designed for 24 or 96 well plates.”

UMAPS:

A reason the UMAPs look different is the size of the image being analyzed creates inherent artifacts in the vector embedding within the analysis underlying the UMAPs, which also underlies why distinct machine learning models were needed. In other words, the field of view and pixels are a variable in the UMAP computation and influence the ‘shape’. We allude to these differences in the second results section with the text: “We trained two separate random forest classifier (RFC) models for each 24-well plate tracking data and 96-well plate tracking data, which we found was necessary due to differences in hardware video acquisition and pixel density for 24-well plate and 96-well plate tracking data.”

Pros/cons of wellplates:

In response to the first component, a new section was added to the Conclusions to address the differences among well plate sizes and our results. Starting at Line 629.

Other studies:

Added citations and explanation to results with another study that compared behaviors among well plate sizes. Starting at Line 636.

80. Wolter ME, Svoboda KR. Doing the locomotion: Insights and potential pitfalls associated with using locomotor activity as a readout of the circadian rhythm in larval zebrafish. Journal of Neuroscience Methods. 2020;330: 108465. doi:10.1016/j.jneumeth.2019.108465

Citation 80 was added which concluded that 48 or 96 well plates can inhibit locomotor responses and influence circadian rhythm due to the decreased area zebrafish have. They highlight that longer recordings can impose stress on zebrafish leading to abnormal circadian behaviors.

82. Lovin LM, Scarlett KR, Henke AN, Sims JL, Brooks BW. Experimental arena size alters larval zebrafish photolocomotor behaviors and influences bioactivity responses to a model neurostimulant. Environment International. 2023;177: 107995. doi:10.1016/j.envint.2023.107995

Citation 82 was included as a review which compared an array of well plate sizes in how they alter specific zebrafish behaviors. Their data shows that smaller well diameters can lead to abnormal behavior which is attributed to smaller well sizes.

• The authors should show the distributions of response types in Figure 2A and 2B. How many “VsLBs” were called following acoustic stimuli and vice versa? Latency distributions would also be helpful here, as the known distinctions in latency between acoustic and visual startle responses might help to correctly reclassify some “visual” responses to acoustic stimuli and vice versa. The authors replicate this distinction in Figure 1E vs K, so this should be feasible. The crossover in response types between AsLBs and VsLBs, despite the “like” in the acronym, makes these labels somewhat misleading. Perhaps using labels that classify them based more turn angle, speed, and/or latency would be more appropriate.

Thanks for the suggestions. As Figures have been switched to Sina graphs (per earlier suggestion) the distributions of response types can now be observed for 3B-C for acoustic stimuli and 3E-F for visual stimuli in the 24 well plates along with supplemental Figure 3 B-C & E-F for 96 well plates. We also included the percentage observed of each response in the legends. From this, it stands out in the data that ‘non-expected’ responses (e.g. visual during acoustic stimuli), are distinct based on frequency and motor parameters (heading and/or speed).

In the conclusions (Lines starting at 658) we note that trained machine learning models have limitations and could (for example) misclassify large turns as startle-like responses, which something comparable may be happening here and is part of our justification for using the ‘like’ descriptor with startle responses. We are open to further changes, yet our aim has been to present a transparent overview of the outputs of our models versus guidance for how to organize data or follow-up analysis based on individual user goals.

The current data shown was also filtered based on latency as acoustic calls were made within 5-5.25 seconds of the stimulus and visual from 5-6 seconds as noted in the legend. If the reviewer believes it would be beneficial, we could refine the visual window from 5.25-6 seconds? Refinement beyond 0.25 second intervals is not possible due to our sliding window analysis, which these limitations we have addressed in the conclusions (Starting at Line 658). We appreciate further classification refinement may be possible with angle, speed, etc, yet a primary goal for this work was to develop machine learning models that take supervised data to create a more agnostic approach to imposing motor calls on data, which these models are based on highly dynamic changes across the pose-estimation based key points that likely include angular, speed, etc. metrics. Unfortunately, the weighting of such individual metrics cannot be readily extracted from such machine learning models.

• Figure 4C,D: it seems that the n’s are defined as responses, but perhaps the more relevant biological unit is the fish. Are some of these differences in heading and speed driven by individual larvae rather than shared changes happening across the population that was tested? This question cannot be answered the way the data is presented.

N’s are defined as fish for these panels, however we can’t ensure that every fish engages in every type of behavior we classified. To make this clearer, we added the total fish tested for each group to the Fig 4 legend. For example, “Control Fish N=144, Behavior: Scoot: N=65, Turn: N=123, AsLB: N=75, VsLB: N=54” indicates we tested 144 fish and 123 of them engaged in a turn during the stimulus window across 6 trials.

• Do the drug treatments in Figure 4 change the distribution of response types? It would help to show something like Figure 5B.

We added additional panels to Supplemental Figure 4 showing the same data as Figure 5B. See Supplemental Figure 4 A-B, E-F. Supplemental Figure 4 legend was updated to reflect these changes.

• The authors’ rationale for the 4-AP and muscimol experiments is to confirm that the MCAM and their deep learning model can detect the behavioral c

---

## [Decision Letter · Decision Letter 1]

7 Oct 2025

High throughput machine learning pipeline to characterize larval zebrafish motor behavior

PONE-D-25-34999R1

Dear Dr. Horstick,

We’re pleased to inform you that your manuscript has been judged scientifically suitable for publication and will be formally accepted for publication once it meets all outstanding technical requirements.

Kind regards,

Yevgenya Grinblat, Ph. D.

Academic Editor

PLOS ONE

Additional Editor Comments (optional):

Reviewers' comments:

Reviewer's Responses to Questions

**Comments to the Author**

1. If the authors have adequately addressed your comments raised in a previous round of review and you feel that this manuscript is now acceptable for publication, you may indicate that here to bypass the “Comments to the Author” section, enter your conflict of interest statement in the “Confidential to Editor” section, and submit your "Accept" recommendation.

Reviewer #1: All comments have been addressed

Reviewer #2: All comments have been addressed

2. Is the manuscript technically sound, and do the data support the conclusions?

Reviewer #1: Yes

Reviewer #2: Yes

3. Has the statistical analysis been performed appropriately and rigorously? 

Reviewer #1: Yes

Reviewer #2: Yes

4. Have the authors made all data underlying the findings in their manuscript fully available?

Reviewer #1: Yes

Reviewer #2: Yes

5. Is the manuscript presented in an intelligible fashion and written in standard English?

Reviewer #1: Yes

Reviewer #2: Yes

6. Review Comments to the Author

Reviewer #1: The authors have thoroughly addressed all of the concerns raised. I strongly endorse the publication of this manuscript.

Reviewer #2: The authors have addressed each of my concerns appropriately, and the manuscript provides a significant addition to the field through the creation and validation of the MCAM and behavioral analysis tools. There are still a couple of typos that remained uncorrected that readers may find confusing: Line 100: “Tupfel long fin” not “Tubingen long fin"; and Line 66: “throughput” not “throughout”.

7. PLOS authors have the option to publish the peer review history of their article (what does this mean? ). If published, this will include your full peer review and any attached files.

**Do you want your identity to be public for this peer review?** For information about this choice, including consent withdrawal, please see our Privacy Policy .

Reviewer #1: No

Reviewer #2: No

---

## [Editor Report · Acceptance letter]

PONE-D-25-34999R1

PLOS ONE

Dear Dr. Horstick,

I'm pleased to inform you that your manuscript has been deemed suitable for publication in PLOS ONE. Congratulations! Your manuscript is now being handed over to our production team.

Kind regards,

on behalf of

Dr. Yevgenya Grinblat

Academic Editor

PLOS ONE